# Efficacy of a Chitin-Based Water-Soluble Derivative in Inducing *Purpureocillium lilacinum* against Nematode Disease (*Meloidogyne incognita*)

**DOI:** 10.3390/ijms22136870

**Published:** 2021-06-26

**Authors:** Jiang Zhan, Yukun Qin, Kun Gao, Zhaoqian Fan, Linsong Wang, Ronge Xing, Song Liu, Pengcheng Li

**Affiliations:** 1CAS and Shandong Province Key Laboratory of Experimental Marine Biology, Center for Ocean Mega-Science, Institute of Oceanology, Chinese Academy of Sciences, Qingdao 266071, China; zhanjiang9312@163.com (J.Z.); kung1993@163.com (K.G.); fanzhaoqian@qdio.ac.cn (Z.F.); wanglinsong@qdio.ac.cn (L.W.); xingronge@qdio.ac.cn (R.X.); sliu@qdio.ac.cn (S.L.); 2Laboratory for Marine Drugs and Bioproducts, Pilot National Laboratory for Marine Science and Technology (Qingdao), No. 1 Wenhai Road, Qingdao 266237, China; 3University of Chinese Academy of Sciences, Beijing 100049, China

**Keywords:** biological control, plant parasitic nematodes, chitin, *Purpureocillium lilacinum*, 6-oxychitin

## Abstract

Plant-parasitic nematodes cause severe economic losses annually which has been a persistent problem worldwide. As current nematicides are highly toxic, prone to drug resistance, and have poor stability, there is an urgent need to develop safe, efficient, and green strategies. Natural active polysaccharides such as chitin and chitosan with good biocompatibility and biodegradability and inducing plant disease resistance have attracted much attention, but their application is limited due to their poor solubility. Here, we prepared 6-oxychitin with good water solubility by introducing carboxylic acid groups based on retaining the original skeleton of chitin and evaluated its potential for nematode control. The results showed that 6-oxychitin is a better promoter of the nematicidal potential of *Purpureocillium lilacinum* than other water-soluble chitin derivatives. After treatment, the movement of J2s and egg hatching were obviously inhibited. Further plant experiments found that it can destroy the accumulation and invasion of nematodes, and has a growth-promoting effect. Therefore, 6-oxychitin has great application potential in the nematode control area.

## 1. Introduction

Plant-parasitic nematodes (PPNs) are one of the main diseases that affect crop yields and can cause economic losses of up to $157 billion per year [1]. Currently, the most effective measure in controlling nematode diseases is using chemical nematicides. However, the inefficiency and abuse of chemical nematicides, such as methyl bromide (MeBr) and aldicarb [2,3,4], have created many problems, such as high toxicity, high residue, and environmental pollution, which seriously threaten human health and the survival of beneficial biological populations that have gradually been on the verge of being eliminated. Therefore, as a green, safe, and efficient treatment method, biological control has gradually attracted attention. Regarding the biological control of nematodes, most of the current research is on the use of nematode antagonistic microorganisms to control the occurrence of nematode diseases. To date, the biological control of PPNs has relied mainly on the direct application of live spores. However, due to the complexity of the soil, it is difficult for biocontrol microorganisms to colonize the soil, resulting in unsatisfactory control effects that are difficult to apply widely in actual production [5,6].

Chitin (β-1,4-n-acetylglucosamine) is a biopolymer that is widely distributed in nature with excellent biocompatibility, biodegradability, low toxicity, and diverse biological activities. Additionally, chitin is considered a good soil organic amendment that has been widely studied in agriculture due to the fact that it can stimulate the growth of chitinolytic microorganisms and increase their biological control activity to enhance plant protection [7,8,9]. Chitin and its derivatives have been proven to be effective in controlling plant pathogens, including fungi, bacteria, viruses, and nematodes [10,11,12,13]. Chitin is applied to the soil to control nematodes by producing nematicidal substances such as ammonia or regulating the soil microbial community environment. Due to the poor solubility of chitin, some experiments have shown that in order to make beneficial changes in soil microorganisms and achieve a satisfactory control of nematodes, the amount of chitin applied needs to reach 5 tons/hectare, which is obviously too large and expensive [14]. In addition, the excessive use of chitin in practical applications can cause phytotoxicity [15,16]. To solve the shortcomings of the large amount of chitin in the application process and its low effect, some studies have used methods by improving the water solubility of chitin, or combining chitin with biocontrol microorganisms to solve the problem of difficult colonization of biocontrol microorganisms in the soil and have a synergistic effect [17,18]. Studies have found that the combination of chitosan and *P. chlamydosporia* can significantly reduce root-knot nematode diseases. Chitosan not only did not inhibit growth but also promoted spore production and increased the parasitism of *P. chlamydosporia* on eggs [19,20]. Hence, it is speculated that by improving the water solubility of chitin to increase its bioavailability, the dosage and cost of the application might accordingly be reduced, thereby reducing the cost and expanding the application rang of chitin.

At present, chitin is treated by two main methods to improve the solubility, one of which is through removing the acetyl group or reducing the molecular weight to obtain homologous water-soluble derivatives of chitin [21,22]. Another is by introducing exogenous hydrophilic groups on the chitin skeleton, such as carboxymethyl and quaternary ammonium salts [23,24]. In recent years, 6-oxychitin, the specific oxidation product of C-6 of chitin, has gradually attracted increasing attention because of the introduction of a carboxyl group based on maintaining the original basic structure and not changing the physical and chemical properties, which not only improves water solubility but also imparts new physiological properties. In addition, compared with other chemical modifications, the reaction conditions of 6-oxychitin are milder, and the product does not have any residual harmful substances. Moreover, the whole preparation process is relatively environmentally friendly [25]. However, reports on the application of 6-oxychitin have focused mainly on its moisturization and coupling with other macromolecules, and there are few studies reported in agriculture application [26,27]. Compounds containing carboxylic acid groups such as jasmonic acid and salicylic acid are important signaling molecules for plant resistance [28,29]. The introduction of carboxylic acid groups can enhance the ability of chitin to induce disease resistance to a certain extent. In addition, carboxylic acid, polysaccharide polymers, and amino acids are the main regulating factors for the growth of soil microorganisms, which are more easily absorbed and utilized by microbial cells [30,31]. Therefore, 6-oxychitin has a great application potential in the direction of inducing biocontrol microorganisms to control plant nematode diseases.

In this study, to explore the application of 6-oxychitin in the field of nematode control, firstly, we used an efficient and environmentally friendly method to prepare 6-oxychitin and compared it with other water-soluble chitin derivatives for their nematicidal-inducing ability to prove that 6-oxychitin has the best induction effect towards *Purpureocillium lilacinum*. Next, we further verified the effects of controlling nematodes by measuring the movement of J2s and egg hatching ability. In addition, we used a soil-free system-gel experiment to determine its control effect on cucumber root-knot nematode diseases and initially explored the mode of action. Finally, we evaluated its safety by measuring the phytotoxicity. This experiment hopes to fill the gap in the application of 6-oxychitin in agriculture and provide a reliable research basis for the development of new safe and green biological nematicides.

## 2. Results

### 2.1. Characterization

With the goal of improving the water solubility of chitin, we first conducted a synthesis experiment of 6-oxychitin and performed a series of characterizations of its structure and properties. As shown in Figure 1A, the infrared spectrum of 6-oxychitin is basically the same as the infrared spectrum of chitin, because 6-oxychitin introduces carboxyl groups without changing the backbone structure of chitin. The infrared spectrum of 6-oxychitin gave birth to two new weak peaks at 1651 cm^−1^ and 1259 cm^−1^, which were attributed to the vibration peaks of the -C=O- and -C-O- of the carboxyl group, respectively. The 6-oxychitin was prepared by oxidizing the primary hydroxyl groups of chitin C-6 to carboxyl groups, so its C6 signal should be weakened, and a new carbonyl peak appears at 178 ppm. Figure 1B accurately verified this conversion. Under a scanning electron microscope, we observed that the internal hydrogen bond formation ability of chitin after TEMPO/NaBr/NaClO treatment decreased sharply, and the structure was severely damaged. Therefore, the structure of 6-oxychitin was looser, and the surface fibrous structure was also reduced (Appendix A). The water solubility test found that the solubility was greatly increased, and the solubility was 26 mg/mL (Appendix A). In conclusion, the target compound was successfully synthesized and has good water solubility.

### 2.2. Screening for Nematicidal Microorganisms Utilizing Chitin

To determine whether chitin can enhance the biological control activity of nematicidal microorganisms and to screen out microorganisms that have synergistic effects with chitin, several common nematicidal microorganisms were used as experimental objects, and the nematicidal activity was measured in vitro. As shown in Table 1, the selected biocontrol microorganisms had a certain level of nematicidal activity, and the J2s mortality of culture filtrates (diluted 1X) at 48 h was approximately 50%, except for *B. subtilis.* When chitin was used as the only carbon source for fermentation culture, the nematicidal activity of the culture filtrates changed. The nematicidal activity of *Purpureocillium lilacinum* and *B. subtilis* was increased by 18% and 13%, respectively, after 48 h. The nematicidal activity of other microorganisms decreased slightly. Therefore, the experiments found that chitin is not synergistic for each biocontrol microorganism, but chitin can obviously improve the nematicidal activity of *Purpureocillium lilacinum* in vitro.

### 2.3. Screening of the Optimum Water-Soluble Chitin Derivatives

Although chitin can enhance the nematicidal activity of *Purpureocillium lilacinum*, the increase is limited. We speculate that this limited increase may be because of the low solubility of chitin, which leads to the low utilization of microorganisms. Therefore, in order to solve this problem, five water-soluble chitin derivatives were selected to find the best chitin substitute. As shown in Table 2, 6-oxychitin dramatically enhanced the nematicidal activity of *Purpureocillium lilacinum* and increased this nematicidal activity by 43% compared with the untreated group. Therefore, 6-oxychitin is the most useful chitin substitute.

### 2.4. Screening the Factors Affecting Nematicidal Activity

We found that the nematicidal activity of *Purpureocillium lilacinum* was greatly improved after 6-oxychitin induction. Therefore, to achieve better control effects, the experiment was carried out by measuring factors such as initial pH, the concentration of 6-oxychitin, spore inoculum, and fermentation time on the nematicidal activity to screen out the most suitable culture conditions. Figure 2a shows that after 24 h of treatment, when the pH was 7, the nematicidal activity of the culture filtrates of *Purpureocillium lilacinum* combined with 6-oxychitin (OTC-PLF) was the highest, and the corrected mortality rate was 81.72%. At 48 h, the nematicidal activity with an initial pH values of 6, 7, and 8 all reached 100%. When the concentration of 6-oxychitin was 1% and 2%, the nematicidal activity reached 100% at 48 h (Figure 2b). Figure 2c shows that when the inoculation amount was 10^5^ spores/mL, the highest nematicidal activity was observed. With a prolonged fermentation time, the nematicidal activity of OTC-PLF gradually increased, and the activity was highest on the 10th day (Figure 2d). In summary, to enhance the nematicidal activity, the best fermentation conditions of OTC-PLF were 1% concentration, 10^5^ spores/mL inoculation amount, 7 for initial pH, and 10 d of fermentation time. The pH change of the fermentation broth will also affect its nematicidal activity, as shown in Figure 2e. The nematicidal efficacy of OTC-PLF at pH 8 and 9 was statistically the same, killing 86.49% and 88.69% of *M. incognita* larvae at 48 h respectively, which was similar to the activity of untreated OTC-PLF. The nematicidal activity is greatly decreased at a low pH.

### 2.5. Anti-Nematode Assay

The preliminary screening test proved that adding 6-oxychitin can enhance the nematicidal activity of *Purpureocillium lilacinum*. To verify its effect, this experiment explored its possible mode of action in vitro, mainly determining its effect on the locomotion of J2s and egg hatching. As shown in Figure 3a,b, treatment with the culture filtrate of *Purpureocillium lilacinum* (PLF) or OTC-PLF significantly decreased locomotion ability compared with the control treatment (Figure 3c), which indicated that the culture filtrates were toxic to nematodes. The nematode was in a frozen state (as shown in Figure 3d). However, the nematodes after 6 h of treatment with OTC-PLC did not die, but the frequency of body bending was less than 1 per minute, and after 12 h of treatment, the head almost no longer thrashed.

The number of larvae was recorded to investigate the effect of culture filtrates on the egg hatching of *M. incognita*. As shown in Figure 4a, there were significant differences in the numbers of hatched larvae with or without OTC-PLF or PLF treatment, which indicated that the toxic effect of culture filtrates reflected the ability to inhibit the egg hatching of *M. incognita*. After 7 days of OTC-PLF treatment, the egg hatching rate was only 17%, which was reduced by 20% compared to the PLF treatment group. These results showed that OTC-PLC has more toxicity than PLC treatment. Under the microscope, the eggs treated with OTC-PLC developed abnormally compared with the normal eggs (Figure 4b), and the toxic effect was reflected mainly in the egg hatching or early development of *M. incognita*. Problems such as embryo damage and eggshell rupture occurred in the eggs, which prevented the larvae from hatching normally (as shown in Figure 4c).

Observation under the scanning electron microscope showed that the body wall of the nematodes in the control group was smooth and full, while the body walls of the nematodes after OTC-PLC treatment showed obvious shrinkage, and the nematodes were more likely to break after being immobilized (Figure 5a,b). As shown in Figure 5c,d, the eggs in the treatment group were full without damage, but the eggshells after OTC-PLF treatment were prone to rupture, and thinner, and the larvae in the eggs developed abnormally.

### 2.6. Effect of OTC-PLF on M. Incognita in Cucumber

#### 2.6.1. Effect of OTC-PLF on Nematode Infecting Cucumber

Root-knot nematodes infect plants and cause their roots to swell, forming typical nodular root knots, which is an important diagnostic feature. This experiment used Pluronic gel to simulate natural soil conditions to preliminarily determine the effect of culture filtrates on nematode infestation in plants. As shown in Figure 6, the effect was concentration dependent. As the concentration increased, the inhibition rate of the root knots also gradually increased. In addition, the inhibition effect of the OTC-PLF treatment group was significantly higher than the inhibitory effect of the PLF group. When the concentration of OTC-PLF treatment was 10%, the relative inhibition rate was as high as 99%, increasing by 20% compared with the PLF treatment. These data suggested that OLC-PLF markedly decreased the infection of *M. incognita*, which further illustrated that the addition of 6-oxychitin has an obvious induction effect.

#### 2.6.2. Effect on the Invasion of Root-Knot Nematode

In the early stage of the experiment, the control effect of OTC-PLF on nematodes was verified in vitro and in vivo. Therefore, to further explore its control mechanism, the invasion process of OTC-PLF was preliminarily observed. As shown in Figure 7a,e, numerous nematodes gathered around the roots of the control group, and the number increased with time. However, the number of nematodes around the roots after treatment was significantly reduced. After OTC-PLF treatment, the nematodes around roots were significantly lower than those of the PLF treatment group (Figure 7c,d). Tomato roots were stained with acid fuchsin at 24 h and 48 h to observe the nematodes were present inside the root. As shown in Figure 7b, nematodes in the roots invaded the plant roots at 24 h, and the invasion rate in the control group reached over 60% at 48 h. The number of root nematodes in the treatment group was much lower than the number of root nematodes in the control group, especially the OTC-PLF treatment group, which had almost no nematodes in the roots (Figure 7f–h). In summary, *Purpureocillium lilacinum* culture filtrates can affect the stage of nematode invasion of plant roots, and the effect is more significant after adding 6-oxychitin.

### 2.7. Phytotoxicity Assay

Considering the potential effect of OTC-PLF as a nematicide in soil, it is necessary to study its potential toxicity to plants. Therefore, the effect of PLF treatment on cucumber seed germination and root length was measured. As shown in Table 3, the OTC-PLF treatment enhanced seed germination and root growth, which was more effective at high concentrations. While the development of seeds was inhibited to a certain extent after PLF treatment, and the germination rate was significantly lower than the germination rate of the control. The GI values were greater than 100% when the concentration of OTC-PLF was between 2.5% and 20%, which showed that seed germination was stimulated to a certain extent. In contrast, the GI value of PLF at a 20% concentration was lower than 80%, which implied that PLF has an inhibiting effect on cucumber seed germination. The RGI values after treatment were all between 0.8 and 1.2, which indicated no significant effect on root growth. In summary, OTC-PLF treatment can promote the growth of plants to a certain extent and can alleviate the phytotoxicity caused by treatment with a high concentration of PLF.

## 3. Discussion

The large-scale use of chemical pesticides has brought about many problems, such as high toxicity and high residues, and has seriously hindered the sustainable development of agriculture. Therefore, significant efforts are being made to exploit safe and effective green natural compounds. As an important biological resource in the ocean, chitin is widely used in agriculture because of its large reserves, lack of toxicity, good biocompatibility, and broad-spectrum antibacterial activity. Previous studies have shown that chitin as a soil amendment can effectively alleviate root-knot nematode disease. The reason is theorized to be that chitin amendment can adjust the soil microbial community structure, which increases the population of chitinolytic microorganisms and other potentially antagonistic microorganisms and eventually improves the control effect on nematodes [32]. Therefore, chitin amendment is more conductive to controlling nematode diseases by selecting suitable biocontrol microorganisms that have a synergistic effect with chitin. In view of the complex soil environment and numerous types of microorganisms, our study selected several biocontrol microorganisms commonly reported in the literature, which included mainly fungi such as *Paecilomyces* sp., and *Aspergillus* sp., bacteria such as *Bacillus* sp., and actinomycetes such as *Streptomyces sp*. [33,34,35,36], and measured the nematicidal-induction effect of chitin on these biocontrol microorganisms to preliminarily observe whether chitin can be used by biocontrol microorganisms. The nematicidal ability of different biological control microorganisms is different in response to the addition of chitin, and not all their nematicidal effects are enhanced. As the only carbon source, chitin has different effects on the growth process of different microorganisms, which indirectly affect the production of secondary metabolites and subsequent biological control activities. The *Purpureocillium lilacinum* showed the best utilization of chitin, and its nematicidal ability increased by 18% at 48 h. *Purpureocillium lilacinum* combined with chitin amendments in the soil reduced the number of root galls, and juveniles of *M. arenaria* have also has been proven [37]. This indicated that chitin could stimulate the production of nematicidal metabolites of *Purpureocillium lilacinum* to a certain extent.

Studies have found that adding 100 g and 200 g of chitin to 150 cm^3^ soil infected by *M. incognita* significantly reduced the number of galls and nematode eggs [38]. However, due to the poor water solubility of chitin resulting in low bioavailability, shortcomings such as a large dosage and slow effect ensue, severely restricting the further application of chitin. To solve this problem, we tried to improve the water solubility of chitin to enhance its utilization by *Purpureocillium lilacinum*, thereby further improving its nematicidal activity. The results of the experiment found that the selected water-soluble derivatives of chitin all enhanced the nematicidal ability of *Purpureocillium lilacinum* to vary degrees. Therefore, improving the water solubility of chitin can enhance its bioavailability to some degree. However, the study also found that increasing the water solubility of chitin is not the main determinant of enhancing its activity. The enhancement of activity may also be related to its skeleton and active group. Among these factors, the induction effect of 6-oxychitin is the most significant, which is good proof of the enhancement of activity. Our results showed that the introduction of groups with inducing function can further enhance the inducing activity based on retaining the original function of chitin, providing a certain research path for future chitin structural modification. In the early days, our laboratory had similar reports that the introduction of chitosan into the carboxyl group can significantly increase the induced resistance of plants to pathogenic fungi, stimulate the production of resistance factors, and regulate and promote plant growth [39].

*Purpureocillium lilacinum*, a very promising nematophagous fungus, is known to secrete certain active compounds that possess larvicidal and ovicidal properties. 6-Oxychitin can induce or enhance the ability to produce nematicidal active compounds to a certain extent. Our results proved that the induction process of active compounds is related not only to the type of carbon source but also to the fermentation conditions. Microbial metabolism can be influenced by pH, inoculation amount, carbon source concentration, and fermentation days. Adjusting these variables can change nematicidal activity. Nematicidal compounds can be stably produced in the pH range of 5–9, but the most suitable pH is neutral from the perspective of 24 h. Although the biomass of *Purpureocillium lilacinum* has been reported to be the largest at pH 6 [40], this condition may not be suitable for the production of nematicidal compounds. Good growth does not mean that it produces more active substances [41]. Optimization experiments of inoculum amount, carbon source concentration, and fermentation days all verified this observation. In addition, to initially explore the properties of the active substances in the culture filtrate, the acid-base stability of the active substances was studied. Under acidic conditions, the nematicidal activity is significantly reduced. The acid-labile substances may play a major role. Although previous reports showed that the nematicidal activity of *Purpureocillium lilacinum* under the induction of karanja cake medium decreased under alkaline conditions [42], which means that new nematicidal compounds may be produced after 6-oxychitin induction, this finding may provide a clue to the development of novel nematicidal compounds.

The culture filtrates of *Purpureocillium lilacinum* induced by 6-oxychitin (OTC-PLF) improved the nematicidal effect compared with the culture filtrate of *Purpureocillium lilacinum* (PLF) treatment, which is mainly reflected in two main aspects: direct killing and control of nematode invasion. After treatment with OTC-PLF, the locomotion and egg hatching abilities of nematodes were decreased. The decrease in the frequency of head and body movements of nematodes was probably due to the toxic substances acting on their motor nerve, slowing their movement [43]. The reduction in egg hatching ability may be due to the active compounds of culture filtrates severely damaging the embryo development of *M. incognita*. In addition, the body wall of nematodes was crumpled and the eggshell was broken as observed under a scanning electron microscope, which may be caused by chitinase and protease in the culture filtrates [44]. The antagonistic effects of OTC-PLF treatment against *M. incognita* were also evident from observations on nematode infection in cucumber. The J2s that penetrated cucumber roots were consistently lower in PLF, possible because the mobility and invasion capabilities of nematodes were impaired. Furthermore, OTC-PLF can promote the growth of plants to a certain extent and can alleviate the phytotoxicity caused by treatment with a high concentration of PLF. In conclusion, 6-oxychitin has great potential to replace chitin as a soil amendment. However, it is evident that practical nematicidal activity needs to be tested in the field, and the specific induction mechanism of 6-oxychitin still needs to be further explored.

## 4. Materials and Methods

### 4.1. Materials

Several common soil biocontrol strains were selected for a strain screening test: *Purpureocillium lilacinum* (BNCC 336533), *Bacillus licheniformis* (BNCC 189067), *Bacillus subtilis* (BNCC 188062), *Streptomyces roseofulvus* (BNCC 152535), and *Streptomyces cuspidosporus* (BNCC153306), which were purchased from BeiNa Culture Collection; *Aspergillus* sp. (laboratory self-screening). Chitin was purchased from Shanghai Aladdin Biochemical Polytron Technologies Inc. The chitin oligosaccharides were bought from TOKYO CHEMICAL INDUSTRY CO. LTD. Chitosan was obtained from Qingdao Yunzhou Biochemical Corp. (Qingdao, China). Chitosan and chitosan citrate (CSC) of different molecular weights were obtained from our laboratory. The chemicals used in the experiments were of analytical grade.

### 4.2. Synthesis and Characterization of C-6 Oxidized Chitin (6-Oxychitin)

#### 4.2.1. Preparation of the C-6 Oxidized chitin (6-Oxychitin)

6-Oxychitin was prepared mainly following the the method of Gao et al. and Muzzarelli et al. [25,39], and was slightly modified. Chitin (1 g) was dissolved in 50 mL DMAc-LiCl (5%, *v*/*v*), and then reprecipitated with water to obtain amorphous chitin. TEMPO (0.24 g) and NaBr (0.4 g) were added to an aqueous suspension of chitin (1 g dry weight / 50 g water), followed by NaOCl (24 mL, 4%). Then adjust the pH to 10.8 with the concentration of 1 M HCl and 0.4 M NaOH was used to maintain the pH of the reaction system at 10.8 for 30 min. Lastly, the reaction was adjusted to neutral with drops of 0.4 M HCl. The mixture was precipitated with a 3–5 times volume of ethanol for 24 h at 4 °C, centrifuged to collect the precipitate, dialyzed, and dried. The synthesis process is shown in Scheme 1.

#### 4.2.2. Characterization and Analytical Method

The Fourier Transform Infrared spectrum (FTIR) obtained using a Thermo Scientific Nicolet iS10 spectrometer ranges from 4000 cm^−1^ to 400 cm^−1^. ^13^C NMR spectra were recorded with Bruker AVANCE III 600 M in solid form. The morphology characteristics were photographed by scanning electron microscopy (Hitachi-3400N). The average molecular weight was estimated by High Performance Liquid Chromatography (Agilent Technologies, USA) equipped with a refractive index (RI) detector, and chromatography was performed on TSK G3000- PWXL columns. The water solubility was measured by the method of Chivangkul et al. [45]. The oxidative degree and carboxylate content were determined according to the methods reported by Pierre et al. [46].

### 4.3. Culturing and Collecting of Nematode

A pure culture of the root-knot nematode *M. incognita* was maintained on tomato (Zhongshu No. 4) in a phytotron, and the culturing method was performed in pots and trays according to the protocol of Atamian et al. [47]. When the nematodes had been inoculated for approximately 40 days, the eggs of the nematode were extracted from infected roots according to the scheme of Fan et al. [48]. The collected eggs were used immediately for hatching in a double-layered paper put in a Petri dish containing distilled water at 28 °C for approximately a week under dark conditions to obtain J2s. Freshly collected eggs and hatched J2s were used for the subsequent experiments. The collected eggs and J2s were used immediately without storage.

### 4.4. Screening for Nematicidal Microorganisms Utilizing Chitin

To screen out the strains that have synergistic effects with chitin, several common soil biocontrol microorganism strains were used as the experimental object, which were screened by adding chitin into the medium and then determining its nematicidal activity in vitro. The studies performed in liquid media were carried out in a modified mineral-based medium (MM) containing 1 g/L KH_2_PO_4_, 2 g/L Na_2_HPO_4_, 0.3 g/L NaCl, 3 g/L KNO_3_, 0.3 g/L MgSO_4_.7H_2_O, 0.01 g/L FeSO_4__,_ and 3 g/L tryptone. The chitin at a concentration of 10 g/L was added to MM. The medium pH was adjusted to 7.0 before autoclaving using NaOH or HCl solution. Subsequently, fungal spore suspension (2 × 10^5^ spores/mL) and bacterial culture medium (2.5 × 10^8^ CFU/mL) were mixed with the medium, and then were cultured at 28 °C for 10 days at 200 rpm/min. The culture medium was centrifuged at 10,000 rpm for 10 min, filtered through a 0.22 μM filter membrane, and stored in a −20°C refrigerator for the nematicidal test in vitro.

To determine the nematicidal activity, 380 μL culture filtrates (diluted 1X by sterile water) and 20 μL J2 suspensions of *M. incognita* containing 50 hatched J2s were placed in a the 48-well plate. The MM liquid medium without microorganisms and samples added served as a blank control. In addition, the J2s were placed in dark conditions at 28 °C. After 24 h, and 48 h, the numbers of dead juveniles were counted, and the mean percentage mortality was calculated. If the body was stiff and could not bend when touched by needles or stimulated with 4% (w/v) NaCl solution, the nematode J2s were classified as dead. Each treatment had four replicates and was repeated three times.

Juveniles’ mortality was calculated as followed:(1)$$\mathrm{Corrected}\;\mathrm{mortality}\;(\%)=\frac{\mathrm{mortality}\;(\%)\;\mathrm{in}\;\mathrm{treatment}-\mathrm{mortality}\;(\%)\;\mathrm{in}\;\mathrm{blank}\;\mathrm{control}}{100-\mathrm{mortality}\;(\%)\;\mathrm{in}\;\mathrm{blank}\;\mathrm{control}}\times 100$$

### 4.5. Screening of the Optimum Water-Soluble Chitin Derivatives

*Purpureocillium lilacinum* was selected as the test strain for the next series of experiments. Different water-soluble chitin derivatives (chitin, 6-oxychitin, chitin oligosaccharides, chitosan, and chitosan citrate) were separately added to MM at a concentration of 10 g/L. The procedures for the preparation of culture filtrates and assessment of nematicidal activity were the same as outlined above. The MM liquid medium without microorganisms and samples added served as a blank control. Each treatment had four replicates and was repeated three times.

### 4.6. Screening the Factors Affecting Nematicidal Activity

6-Oxychitin was selected as the next factor affecting nematicidal activity screening studies. In these experiments, the initial pH (5, 6, 7, 8 or 9), which was adjusted before autoclaving using NaOH or HCl solution, concentration of 6-oxychitin (0.1, 1, 5, 10 or 20 g/L), spore inoculum (10^2^, 10^3^, 10^4^, 10^5^ or 10^6^ spores/mL) and fermentation time (2, 4, 6, 8 or 10 d) effect on nematicidal activity were studied through single-factor experiments [49]. The fermentation methods and nematicidal activity were the same as in Section 4.4 but, if necessary, replaced with a single factor that was tested. The MM liquid medium without microorganisms and samples added served as a blank control. In addition, the effect of pH on nematicidal efficacy of the culture filtrate was also estimated. Briefly, the pH of culture filtrates before nematicidal assays was adjusted to pH 5, 7, 8, and 9 using NaOH or HCl solution. After standing for 1 h, the solution was adjusted back to the original pH (pH = 8), and then filtrates of various pH values were subjected to bioassays. MM liquid medium was used as a blank control, and the corresponding pH buffer was used as a control. Each treatment had four replicates and was repeated three times.

### 4.7. Anti-Nematode Assay

#### 4.7.1. Effect on J2s Locomotion Behaviour

For the locomotion behavior assay, the culture filtrates of *Purpureocillium lilacinum* induced by 6-oxychitin (OTC-PLF) were applied to the J2s. The method was performed mainly according to Tsalik and Hobert [50]. Briefly, 380 μL culture filtrates and 20 μL J2 suspensions of *M. incognita* containing 50 hatched J2s were placed in a 48-well plate. The head thrashing frequency and body bent frequency were recorded at 1, 3, 6, and 12 h post-treatment assisted by a dissecting microscope (45×). The head thrashing and body bend frequency were recorded according to a previous method [51]. The MM liquid medium without microorganisms and 6-oxychitin served as a control. Each treatment recorded the locomotion behavior of 10 J2s and was repeated 3 times.

#### 4.7.2. Egg Hatching Assay

The method was performed mainly according to Roberts et al. [52], and was slightly modified. The collected eggs newly extracted from plants were rinsed 3 times with sterile water. Subsequently, 380 μL culture filtrates and 20 μL egg suspensions (50 eggs) were added into the wells of a 48-well plate. MM liquid medium without microorganisms and 6-oxychitin served as a control. The plate was covered with plastic wrap to prevent evaporation and then incubated at 28 °C. The number of juveniles was recorded at 1, 3, and 7 d post-treatment. The whole experiment was carried out under aseptic conditions. Each treatment had four replicates and was repeated three times.

The rate of egg hatching was calculated as follows:(2)$$\mathrm{Egg}\;\mathrm{hatching}\;\mathrm{rate}\;(\%)=\frac{\;\mathrm{Juveniles}}{\mathrm{Juveniles}+\mathrm{Eggs}}\times 100$$

#### 4.7.3. Effect on Nematode Morphology

The morphology of nematodes, including J2s and eggs, was observed by light microscope and scanning electron microscope. In this experiment, the collected eggs newly extracted from plants and hatched J2s after one week of incubation were rinsed 3 times with sterile water. Subsequently, 380 μL culture filtrates and 20 μL egg suspensions (100 eggs) or 20 μL J2 suspensions (100 J2s) were added to a 1.5 mL sterile centrifuge tube. The J2s were collected on the second day, and eggs were collected on the 7th day after treatment with OTC-PLF for morphological observation. MM liquid medium without microorganisms and 6-oxychitin added served as a control. SEM samples were prepared according to the method of Abolafia et al. [53]. Briefly, the collected J2s and eggs were washed with PBS (0.1 M) 3 times, fixed in 2.5% glutaraldehyde for approximately 2 h, cleaned with PBS several times to remove glutaraldehyde, dehydrated with an ethanol solution of different concentration gradients in turn, dried and sliced for SEM observation. The controlled J2s were killed with gentle heating (temperature controlled at 60–65 °C), heat treated for 3 min, moved into the refrigerator, and stored for approximately 2 h before fixation.

### 4.8. Effect of OTC-PLF on Nematode Infecting Cucumber

#### 4.8.1. Plant Materials and Pluronic Gel Preparation

*Cucumis sativus* ’Jinyan No.4’ was used in this experiment. Before germination, seeds were treated with 10% (w/v) trisodium phosphate for 30 min to sterilize, followed by washing with sterile water several times, and soaking in sterile water for approximately 3 h. Finally, the sterilized seeds were placed on a Petri dish covered with wet Whatman paper and cultivated for 3–4 days at 28 °C in the dark for the next experiment when the lateral roots grew out.

Pluronic F-127 (PF-127) is a copolymer of propylene oxide and ethylene oxide, with negligible toxicity to nematodes or plant tissues. The nematodes suspended in PF-127 gel can move freely in three-dimensional space, which is more similar to their movement in the soil. Therefore, using this system to simulate the natural soil environment can more effectively assess host–pathogen interactions [54]. Based on these properties, the assay used PF-127 as the medium for nematode infection. PF-127 was purchased from Sigma-Aldrich (USA). The preparation was performed according to the protocol of Xing et al. [55]. In brief, 23 g of Pluronic F-127 powder was added to 80 mL of pre-cooled sterile water dissolved under stirring at 4 °C for 24 h, and then stored at 15 °C for later experimental use.

#### 4.8.2. Effect on Root Infection in Pluronic Gel

Pluronic F-127 (PF-127) gel was used as a medium for nematode infestation [55]. The OTC-PLF was diluted with PF-127 to different concentrations (10%, 5%, 2.5%, 1%, 0.1%), and then poured it into each Petri dish (3.5 cm diam.) approximately 3 mL. Seedlings of cucumber with approximately 2 cm long root tips were placed were placed in the center of the Petri dish. Below 15 °C, approximately 50 J2s were inoculated by pipette at a distance of about 2 mm from each root tip used. Then, each Petri dish (3.5 cm diameter) was placed in a humid plate and inoculated in a light incubator at 26/20°C (day/night) with a 16:8 h light: dark photoperiod. Each treatment group contained 7 plates, and each experiment was repeated at least 3 times. After 1 day of inoculation (1 dpi), the seedlings were rinsed with distilled water and transferred to wet filter paper with Hoagland’s nutrient solution. At 7 dpi, the number of root knots was recorded and photographed.
(3)$$\mathrm{Root}\;\mathrm{inhibition}\;\mathrm{rate}\;\left(\%\right)=\frac{\mathrm{Root}\;\mathrm{knot}\;\mathrm{number}\;\mathrm{in}\;\mathrm{treatment}-\mathrm{Root}\;\mathrm{knot}\;\mathrm{number}\;\mathrm{in}\;\mathrm{blank}\;\mathrm{control}}{\mathrm{Root}\;\mathrm{knot}\;\mathrm{number}\;\mathrm{in}\;\mathrm{blank}\;\mathrm{control}}\times 100$$

#### 4.8.3. Effect on the Invasion of Root-Knot Nematodes, *M. Incognita*

Based on the root infection assay, a concentration of 10% culture filtrates was selected to study the process of *M. incognita* invading cucumber roots. The nematode suspension of approximately 200 J2s was mixed thoroughly with 3 mL Pluronic gel containing culture filtrates at 4 °C in a Petri dish (3.5 cm diameter). The germinated seedlings and Petri plates were prepared and incubated as described above. The number of J2s gathered at 2 mm around the root at 4 h and 6 h was recorded. After inoculation at 24 h and 48 h, the plants were stained with acid fuchsin to observe the distribution of nematodes on the roots, pictures were taken and recorded [56]. Each treatment group contained 6 plates, and each experiment was repeated at least 3 times.

### 4.9. Phytotoxicity Assay

The phytotoxicity assay was based on measuring the seed germination rate (germination index, GI), and root length (relative growth index, RGI) as described by Magdaleno et al. [57]. A total of 50 cucumber seeds were treated (10 seeds/Petri dish, five Petri dishes) with different concentrations of culture filtrates (10%, 5%, 2.5%, 1%, 0.5%) and repeated 3 times. Sterile water served as a blank control, and germination was monitored daily. After incubating for 24 h at 28 °C in the dark, the number of germinated seeds was recorded. Then 30 seeds with the same germination potential were selected, and the root length was measured at the end of the experiment (approximately 96 h incubation).

The GI and RGI were calculated by the following equations:(4)$$RGI=\frac{RLS}{RLC}$$
(5)$$GI(\%)=\frac{RLS\times GSS}{RLC\times GSC}\times 100$$
*RLS*: root length of the experimental group;*RLC*: root length of the control group;*GSS*: Germination number of seeds in the experimental group;*GSC*: Germination number of seeds in the experimental group;when *GI* < 80%, this value indicates phytotoxicity, while *GI* > 100%, the value may indicate growth promotion. At the same time, when *RGI* < 0.8, root elongation was inhibited, while when *RGI* > 1.2, root elongation was stimulated.


### 4.10. Statistical Analysis

All the data were analyzed by ANOVA using SPSS version 19.0 software according to Duncan’s method. All results are indicated as the mean ± standard deviation (SD). A *p*-value of less than 0.05 was considered a significant difference.

## 5. Conclusions

In this study, the optimal alternatives of chitin were selected by determining the nematicidal effect of water-soluble chitin derivatives combined with *Purpureocillium lilacinum* and its metabolites. 6-Oxychitin, the C-6 specific oxidation product of chitin, showed the strongest induction effect, which was further verified by in vitro and in vivo experiments and was reflected mainly in the inhibition of nematode movement, egg hatching ability, and nematode infestation in plants. In summary, 6-oxychitin can not only significantly improve the nematicidal ability of *Purpureocillium lilacinum*, but also alleviate the phytotoxicity of chitin and *Purpureocillium lilacinum* alone; thus, 6-oxychitin has broad application prospects in the control of nematode diseases. The study not only expanded the application range of 6-oxychitin but also provided new insights for the prevention and control of agricultural nematode diseases by the combination of organic additives and biocontrol microorganisms and their metabolites.

## Data Availability

Not applicable.

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
