# Peer review of "Efficacy of a Chitin-Based Water-Soluble Derivative in Inducing *Purpureocillium lilacinum* against Nematode Disease (*Meloidogyne incognita*)"

_ijms, 2021, doi:10.3390/ijms22136870_

Round 1

Reviewer 1 Report

Experimental designs should be included. What data were used to prepare figures and charts? Data from the first experiment experimental run, the second run? All three analyze together to create data set presented in your paper. 

Statistical analyses could be improved.

Reviewer 2 Report

This is a research paper reporting on the efficacy of chitin-based water-soluble derivative  and its synergistic effects when combined with Purpureocillium lilacinum (formerly Paecilomyces lilacinus) against the root-knot nematode, Meloidogyne incognita. As stated in the conclusion the reported findings can be useful for the development of new management strategies against these economically important plant parasitic nematodes.

Although the scientific relevance of the work performed is undeniable there are a number of flaws that must be addressed by the authors. The fungus Paecilomyces lilacinus, having undergone a taxonomic revision is correctly named as Purpureocillium lilacinum (see Luangsa-ard et al., 2011 doi: 10.1111/j.1574-6968.2011.02322.x). This needs to be corrected through the entire manuscript.

The English needs thorough revision. The Title should be more focused on what actually is described in the paper. The study evaluates the effects of the chitin-derivative on the nematicidal potential of P. lilacinum. The activity of the chitin-based water-soluble derivative as a nematicide was not assessed.

The Introduction section is long but could be improved as there are some important references regarding the synergistic effects of chitin-based compounds and nematophagous fungus such as Pochonia chlamydosporia (for example: Suarez-Fernandez et al. 2021, doi:10.1111/1462-2920.15408; Escudero et al. 2017, doi: 10.3389/fpls.2017.01415).

The methodologies are described in an abbreviated way and occasionally seem incomplete. For example, it is not clear why other soil biocontrol strains were tested. What do the authors consider freshly collected eggs and hatched juveniles (lines 367, 405)? In some procedures the blank controls and the number of replicates and repetitions is not clearly indicated. How was pH adjusted in test 4.6? Bioassays 4.7.1 and 4.7.3 should be described in more detail specifically on the process of exposing nematode eggs and J2 to the amended culture filtrates? In bioassays 4.8.2 and 4.8.3 the distance between the root tip and the J2 inoculation point is an important detail that is missing.

The Results section would benefit greatly if the order presented in the figures is followed (e.g. 2.4) and if the subsections used in the Materials and Methods component were used (mainly in 2.5). Figure and Table titles and footnotes should be revised. In lines 180-181: the authors did not evaluate the effect of culture filtrates on the nematode reproduction system. They assessed the effects on egg hatching. Also, some statements would be more appropriated if included in the Discussion section (e.g. lines 236-238).

The Discussion is very short and although based on the results obtained it fails into present the major findings in the context of published literature and to recognize the limitations of the present study. This is a main concern that the authors need to address.

Round 2

Reviewer 2 Report

This revised version of the manuscript has major improvements. The effort of the authors to answer the questions raised in the first revision is noticeable. Nonetheless, there are still some details that must be addressed:

Title

The words “nematicidal candidate” must be omitted. As mentioned in the first revision: “The study evaluates the effects of the chitin-derivative on the nematicidal potential of P. lilacinum. The activity of the chitin-based water-soluble derivative as a nematicide was not assessed". This is stated in the manuscript:

  • lines 160-161 “nematicidal activity of the culture filtrates of Purpureocillium lilacinum induced by 6-oxychitin”
  • line 299-300 “We measured whether chitin has a nematicidal-induction effect”

Accordingly, the words “nematicidal induction ability” from the Abstract must also be omitted. Suggestion: Results showed that 6-oxychitin is a better promoter of the nematicidal potential of Purpureocillium lilacinum than other water-soluble chitin derivatives.

Tables 1, 2, 3 footnotes

Please replace with: Different letters represent significant differences (P<0.05) according to Duncan’s test.

Figure 2 and 6 – statistical analysis? significant differences?

For all figures - Please indicate in the figures legends the meaning of OTC-PLF, PLF and CK (where appropriate)

line 413, 414, 466, 474 and 484 – “freshly eggs and J2” – this was not clarified. The authors state on line 413 that the eggs collected from infected roots were stored for hatching “for approximately a week under dark conditions to obtain J2s”. The eggs were used after extraction? And the J2? After one week?

line 519 – “2 cm long root tips were placed” instead of “2 cm long root tips of cucumber were placed”
